# Potential of *Moringa oleifera* to Improve Glucose Control for the Prevention of Diabetes and Related Metabolic Alterations: A Systematic Review of Animal and Human Studies

**DOI:** 10.3390/nu12072050

**Published:** 2020-07-10

**Authors:** Esther Nova, Noemí Redondo-Useros, Rosa M. Martínez-García, Sonia Gómez-Martínez, Ligia E. Díaz-Prieto, Ascensión Marcos

**Affiliations:** 1Immunonutrition Research Group, Department of Metabolism and Nutrition, Institute of Food Science and Technology and Nutrition (ICTAN)—CSIC, C/Jose Antonio Novais 10, 28040 Madrid, Spain; noemi_redondo@ictan.csic.es (N.R.-U.); sgomez@ictan.csic.es (S.G.-M.); ldiaz@ictan.csic.es (L.E.D.-P.); amarcos@ictan.csic.es (A.M.); 2Department of Nursery, Physiotherapy and Occupational Therapy, Faculty of Nursery, University of Castilla-La Mancha, 160071 Cuenca, Spain; rosamaria.martinez@uclm.es

**Keywords:** *Moringa oleifera*, diabetes mellitus, fasting glucose, glucose tolerance, antioxidant enzymes, lipid metabolism, animal studies, human studies

## Abstract

*Moringa oleifera* (MO) is a multipurpose plant consumed as food and known for its medicinal uses, among others. Leaves, seeds and pods are the main parts used as food or food supplements. Nutritionally rich and with a high polyphenol content in the form of phenolic acids, flavonoids and glucosinolates, MO has been shown to exert numerous in vitro activities and in vivo effects, including hypoglycemic activity. A systematic search was carried out in the PubMed database and reference lists on the effects of MO on glucose metabolism. Thirty-three animal studies and eight human studies were included. Water and organic solvent extracts of leaves and, secondly, seeds, have been extensively assayed in animal models, showing the hypoglycemic effect, both under acute conditions and in long-term administrations and also prevention of other metabolic changes and complications associated to the hyperglycemic status. In humans, clinical trials are scarce, with variable designs and testing mainly dry leaf powder alone or mixed with other foods or MO aqueous preparations. Although the reported results are encouraging, especially those from postprandial studies, more human studies are certainly needed with more stringent inclusion criteria and a sufficient number of diabetic or prediabetic subjects. Moreover, trying to quantify the bioactive substances administered with the experimental material tested would facilitate comparison between studies.

## 1. Introduction

The *Moringa oleifera* (MO) tree, known as ‘drumstick tree’ belongs to the Moringaceae family. It is the best known and most widely used of the thirteen species of the *Moringa* genus. It is originally from the southern Himalayas to the north-east of India, Bangladesh, Afghanistan and Pakistan and is nowadays cultivated in tropical and subtropical areas of Africa, America and Asia. It is a fast- growing perennial tree that can measure up to 12 meters in height and displaying great ecological plasticity since it is able to adapt to the most dissimilar conditions of the soil, temperature and precipitation, being very resistant to the drought [1,2]. Its leaves are compound, and are arranged in groups of leaflets with a length of 30–70 cm, the flowers have five unequal white petals and yellow stamens and the fruits in the form of pods contain 12 to 15 winged seeds per fruit and its cultivation is carried out by sowing or using cuttings [3,4].

MO is considered a versatile plant due to its multiple uses. The leaves, green pods and seeds are edible and are used as food as part of the traditional diet in many countries in the tropics and subtropics. The green pods and leaves are consumed as vegetables and boiling is the most widely used method of cooking; the seeds are ground to obtain a flour that is used together with the leaves in the preparation of soups and with other flours (wheat or corn) to make bread and biscuits, improving their nutritional quality [5,6]; the seeds can be consumed fresh or pounded, roasted or pressed into a sweet, high quality oil [7]. The seed oil, due to its thermal and oxidative stability, is used for cooking and as a solidifying agent in the production of margarine and other food products containing solid and semi-solid fats, thus eliminating the process of hydrogenation [8,9]. Although its bitter and a little astringent taste may be a barrier to acceptance, there is currently an increase in consumption of teas prepared from MO leaves in western markets [10].

The flowers, pods, seeds and mainly the leaves are a source of essential nutrients and nutraceuticals [11]. They contain protein, lipids (mainly omega-3 and omega-6 polyunsaturated fatty acids, oleic acid and a low content of saturated fatty acids), carbohydrates, minerals (potassium, calcium, magnesium and iron), vitamins (β-carotene, α-tocopherol and highly bioavailable folic acid) and dietary fiber, and can be a food resource for people who are undernourished [11,12]. On a dry matter basis, the total protein content measured in dry leaf powder ranges from 23 to 35%, which is a high content compared to other local plants of common use [13,14]. Consistent with these data, the analysis of 181 samples from different African and Asian countries showed a mean value of 30.3 ± 4.9% [15]. Total dietary fiber was reported in the range of 20 to 28% by different authors [16,17,18]. The mean concentration of Ca, Cu, Fe, Mg, and Zn in MO leaves collected from a garden in Jalisco State of Mexico were 16100, 9.6, 97.9, 2830 and 29.1 mg kg^−1^ dry weight (dw), respectively [13] and the overall mean concentrations of Ca, Cu, I, Fe, Mg, Se and Zn in MO leaves from different localities in Kenya were 18300, 6.92, 0.218, 202, 5390, 4.25 and 35.6 mg kg^−1^ dw, respectively [19]. The variability is due to genetic background, soil, climate, season and plant age as well as processing and storage procedures. Further, the use of different analytical techniques amplifies the variations [18]. Vitamin E and β-carotene contents were 770 and 185 mg kg^−1^ dw, respectively, in South African MO leaves [16]. Regarding fatty acid profile, α-linoleic acid (omega-3) showed the highest value (44.57%) in this study and the total polyunsaturated fatty-acids (52.21%) were more abundant than saturated fatty acids (43.31%) [16].

MO presents a wide variety of biological activities evidenced in in vitro experiments, showing potent anti-oxidative, analgesic, cytoprotective, anti-ulcer, anti-hypertensive and immunmodulatory actions as well as an inhibitory effect on proinflammatory mediators such as iNOS, COX-2, PGE-2, TNF-α, IL-1β and IL-6 [2,17,20,21,22,23]. It is a popular medicine used in Asia and Africa for multiple purposes, ascribed to the various parts of the plant, which, in addition, are used in many different ways [7]. Its therapeutic value as a cardioprotective, hepatoprotective, neuroprotective, anti-asthmatic, anti-tumor, antimicrobial, hypolipidemic, modulator of intestinal microbiota and anti-diabetic agent derives from its phytochemical constituents such as alkaloids, phenolic compounds (coumarins, tannins, flavonoids) and glycosides (saponins and glucosinolates) although the amount of these metabolites varies according to the geographical location and the extraction method used [7,24,25,26].

The leaf is the most commonly used plant part for therapeutic purposes. The main phytochemical compounds extracted from the leaves of MO include glucosinolates, flavonoids and phenolic acids that have a protective effect against chronic diseases (arterial hypertension, diabetes mellitus, cancer, metabolic syndrome and overall inflammation) [27]. Glucosinolates are highly represented in the Brassicaceae family (cruciferous), belonging, as the Moringaceae family, to the order Brassicales. They are known as mustard oils and are not bioactive until they are hydrolyzed (during the crushing or breakage of the vegetable cells), by endogenous “myrosinase” enzymes resulting in the formation of thiocyanates, isothiocyanates and nitriles which constitute the active molecules with chemo-protective, hypotensive and hypoglycemic effects [7,10,27,28].

On the other hand, the presence of phytates and oxalates, with mineral-binding activity that decreases their absorption and saponins, which show favorable or detrimental nutritional and health effects depending on the ingested amount [29], is to be considered when evaluating the safety of long-term use and the overall guidelines of MO consumption as a functional food or nutraceutical.

Body fat accumulation, insulin resistance development and chronic inflammation are considered the cornerstones of the metabolic alterations leading to diabetes mellitus (DM) type 2. The global diabetes prevalence in 2019 was estimated to be 9.3% (463 million people), rising to 10.2% (578 million) by 2030 and 10.9% (700 million) by 2045 [30]. Treating diabetic patients is one of the highest costs of health care systems and it increases each year, which highlights the relevance of prevention and specifically the level of information and awareness of the prediabetic patient. Five to 10% of prediabetic patients will develop DM each year [31]; hence, identifying and treating prediabetes is relevant to avoid or slow down DM establishment. Pharmacotherapy to control the progress of the disease is not without negative side effects. More drawbacks of drug therapies are long-term loss of efficacy and poor adherence to lifelong treatments [32]. On this basis, plants and herbs with hypoglycemic activity could represent good alternatives, especially for those prediabetic patients who fail to make durable lifestyle changes.

MO has a potent antioxidant activity, which can prevent and protect pancreatic cells from the oxidative stress associated with the hyperglycemic state [33]. This capacity is attributed to the high concentration of polyphenolic compounds such as flavonoids (myricetin, quercetin and kaempferol) and phenolic acids (chlorogenic acid, caffeic acid and the most abundant, gallic acid). On the other hand, the importance of isothiocyanates in glycemic control seems to be related to their ability to reduce resistance to the action of insulin and hepatic gluconeogenesis [28].

The hypoglycemic effect of the MO leaves has also been associated with their fiber content and the presence of flavonoids and phenolic acids through different mechanisms. In vitro experiments have provided evidence that these molecules or the extracts rich in them inhibit the activity of pancreatic α-amylase and intestinal α-glucosidase, decreasing intestinal absorption of glucose and advanced glycation [12,34,35,36,37], thus reducing the risk of developing DM and improving glucose levels in prediabetic and diabetic patients. There are other multiple potential mechanisms involved which include e.g., the inhibition by quercetin glycosides of the Na^+^-dependent glucose uptake via the SGLT-1 transporter [38,39]. These mechanisms are numerous and will not be reviewed here; on the contrary, the purpose of this work is to review the evidence of the potential use of MO for glucose control, both, in animal models and human clinical trials. 

## 2. Materials and Methods

We performed a systematic review of MO effects on the control of blood glucose levels according to PRISMA statement guidelines.

The literature search was performed in PubMed database from August 2019 to April 2020 using the terms “Moringa oleifera” and “diabetes” without year limits. Seventy-three papers were retrieved (Figure 1). Articles were discarded if they were reviews (N = 10) or reported only in vitro cell culture experiments (N = 9) or plant analysis studies (N = 2). All papers covering animal models and human studies were further considered. Among animal studies, six were excluded because they did not present data on glucose control and involved pathologies basically independent from DM or its complications. Among human studies, eight of them were excluded because they were related to plant uses among populations, mainly determined through surveys. This left 38 articles, (35 on animals and three on humans) which were included in Table 1 and Table 2, except for two animal studies, one of them considered methodologically weak and another one testing a synthetic compound, which was not isolated from *M. oleifera*. Regarding human studies, five additional studies not listed in PubMed were retrieved by searching the reference lists in other included articles or in literature reviews.

All animal studies with glucose control related measurements, both in diabetes models and in normal healthy animals are presented in Table 1, which includes also a summary of other evidences related to DM-associated pathophysiological complications. Human studies are presented in Table 2. Finally, the ample set of measurements, from organ to molecular measurements, are systematically classified and further detailed in Table 3 and Table 4. 

## 3. Scientific Evidences of *M. oleifera*’s Effect on Glucose Control in Animal Models

Many studies have investigated the hypoglycemic effect of different parts of MO in animal models, mainly the aqueous or methanolic extracts of leaves and, secondly, seeds. Dry leaves have also been employed as powder. These studies are usually performed in streptozotocin (STZ) or alloxan-induced diabetic rats and a fewer other studies have been performed in obese animals fed high-fat diets (HFD). The three of them are used as models of DM. The first two models rely on chemical destruction of pancreatic β-cells. Both chemicals are employed as cytotoxic glucose analogues that tend to accumulate in pancreatic β cells through glucose transporter 2 (GLUT2). However, depending on the dose of chemical employed, usually through intraperitoneal injection, more resemblance to Type-1 or Type-2 diabetes is observed [77,78]. Moreover, there is some controversy as to whether alloxan and STZ can be used to create type-2 diabetes models since both exert toxicity on β-cells (alloxan through reactive oxygen species [ROS] formation and STZ through DNA methylation that causes a chain of damages leading to DNA fragmentation) instead of inducing insulin resistance, which is the main characteristic in type-2 diabetes. Only at low dose and in combination with HFD [79] or when used with neonatal rats that will develop hyperglycemia in the adult age [36], can STZ be considered to induce type-2 diabetes [77,78]. This is an important point that has not been consistently approached in the experiments performed and the corresponding published works.

In animal models of diabetes, different approaches have been used to administer MO to the animals or incorporate it to the diet. Mainly two ways have been used, either delivery of any type of previously prepared extract in aqueous solution through oral gavage or incorporation of dry material (either lyophilized extracts or blended dry leaves) to the normal diet. As an example, Khan et al. [43] prepared an aqueous extract, obtained through maceration of leaf powder during approximately 24 h, with continuous stirring and then filtration and lyophilization to get a solid residue that is later incorporated to the standard diet.

Another type of procedure is the extraction with organic solvents (*n*-hexane plus methanol, ethanol, etc.) and evaporation to obtain a solid residue, preserved by freeze drying and storage, which is then reconstituted in distilled water previously to the oral administration, commonly through gavage [36,58]. Regarding extract composition, some protocols have been published to standardize extract production in order to control dose and activity of the ingested active components of those extracts [10,56]. These can be helpful techniques to characterize products in nutraceutical development. However, the most frequent form of MO consumption so far, is as a food, mainly fresh or dry leaves, both for animal feeding and human consumption.

### 3.1. Experiments with Raw M. oleifera Leaves or Seeds

Three animal studies have been published investigating the hypoglycemic effect of MO using dry leaf powder. In Villarruel-López et al. study [40], the administration of 50 mg/day MO dry leaf powder during 8 weeks to alloxan-induced diabetic rats led to a decrease in blood glucose, measured by an Accu-Check Active^®^ device (Roche Diagnostics^®^, Indianapolis, IN, USA) at week 2, which tended to remain on later weeks. In this study it is not clear whether glucose was measured in the fasted state since fasting is only explicitly mentioned prior to sacrifice. Similar results were reported by Oboh et al., [41] in STZ-induced diabetic rats who received 2% or 4% MO leaf in their diet for 14 days, with or without acarbose, an α-glucosidase and α-amylase inhibitor. Both doses (2% and 4%) progressively and significantly reduced fasting blood glucose during treatment. In the third study, an acute glucose intolerance ameliorating effect of the MO dry leaf powder was found. This study consisted of an oral glucose tolerance test (OGTT) performed in spontaneously diabetic Goto-Kakizaki rats receiving a single oral dose of glucose (2 g/kg b.w.) plus MO (200 mg/kg b.w.) compared to diabetic rats that received only glucose [39]. This effect was also shown to some extent in normal Wistar rats not suffering from diabetes [39].

In Oboh et al. study [41], when seeds were used instead of leaves, the reduction in fasting blood glucose level was also significant. However, 4% MO leaves in the diet (with or without acarbose) showed the most reducing effect compared to the other leaf (2%) and seed (2% and 4%) supplements. In a different study that also used MO seeds powder, at doses of 50 and 100 mg/kg b.w. mixed with the diet, reductions of 35% and 45% in fasting blood glucose and 13% and 22% in glycated hemoglobin (HbA1C) were observed after a 4-wk treatment compared to the positive control group [42].

### 3.2. Experiments with M. oleifera Aqueous Extract

#### 3.2.1. Acute Effects

Regarding acute hypoglycemic effects, evidence was obtained using leaf aqueous extracts, instead of the powdered leaves, both, with or without oral glucose challenge [43,44]. In the first case, STZ-induced diabetic rats pretreated with 200 mg/kg leaf extract 90 min before an oral glucose challenge showed significantly lower glucose levels at 1, 2 and 3 h post-challenge [44]. In this study, the 100 mg/kg dose had a smaller effect and the 300 mg/kg dose showed a similar effect than the 200 mg/kg dose, which decreased around 25% the glucose values two hours post-glucose. In this respect, other authors using the same diabetes induction model and a dose of 100 mg/kg of the extract observed approximately 50% reduction in 2-h post-challenge blood glucose compared to the control group [43]. Finally, in normal Wistar rats, the hypoglycemic effect during an OGTT performed after 30 min of being gavaged with 20 mL/kg MO leaf tea was studied [45]. An overall decrease of 18% in postprandial glucose was observed in the 150 min following glucose challenge.

The acute anti-hyperglycemic effect was also shown with MO aqueous leaf extract in STZ-diabetic rats which had blood glucose measured (without oral glucose challenge) at 2-hourly intervals after 100 mg/kg extract administration in 2 consecutive days [43] and in normal Wistar rats 6 h after 200 mg/kg extract administration on a single occasion [44]. In Khan et al. study [43], diabetic treated rats showed a maximum fall of 53.2% in fasting glucose after 4 h of oral administration. Similar effects were evidenced in mice whose diabetes was induced through a HFD, showing a 34% decrease in blood glucose on day 2 that normalized compared to control (non-diabetic) mice and more than 50% decrease on day 3 [43].

#### 3.2.2. Long-Term Effects

The MO aqueous leaf extract has also shown a chronic hypoglycemic effect in experiments with long-term treatments. In example, this effect was proven in STZ-diabetic rats along three weeks of intervention with 100 mg/kg [43] or 200 mg/kg [44] aqueous extract compared to STZ-diabetic rats untreated with the extract as well as in HFD-induced diabetic mice, the last ones showing complete normalization in 7 days [43]. The chronic lowering effect of hyperglycemia was also observed in alloxan-induced diabetic rats orally administered MO aqueous extract (250 mg/kg or 300 mg/kg) for 18 days [46,47] or 24 days [48] compared to the blood glucose levels in the control group at the end of the intervention, and in a similar mice model administered with a dose of 100 mg/kg aqueous extract for 14 days [49]. In the last one, both, fasting blood glucose and HOMA-IR were significantly improved compared to diabetic untreated mice. Following similar methods but administering the aqueous MO extract (200 mg/kg) for a longer period that extended for 8 weeks, a 62% reduction in fasting glucose was found in STZ-induced diabetic rats [50], while a lower dose (100 mg/kg) of a similar extract led to a 33% reduction of blood glucose levels over a period of 24 weeks [51]. Further, in a 12-week study, with several OGTT performed over the course of the intervention, the control of blood glucose levels after the oral challenge was confirmed. This C57BL/6J mice model used very high-fat diet with 5% MO leaf concentrate of an aqueous extract providing similar doses of 200 mg MO/kg/day. The area under the curve (AUC) was significantly lower than in the control diabetic animals for the OGTT performed at weeks 8 and 12 (but not at 4th week of treatment) [28]. Mixed findings have been reported with a supplementation (300 mg/kg) lasting 4 weeks in high fructose diet-induced diabetic animals, with a non-significant decrease in glucose (from 133 to 129 mg/dL, mean values) but normalization of hyperinsulinemia (from 5.05 to 2.64 µIU/mL, mean values) [52].

Moreover, experiments have been performed with a leaf protein isolate obtained by aqueous extraction, precipitation and dialysis [53]. When used at concentrations of 300 mg/kg and 500 mg/kg (but not 100 mg/kg), intraperitoneal administration in alloxan-induced diabetic mice significantly decreased blood glucose after a single-dose, as well as after a daily dose for 7 days. However, no effect was shown when administered orally, which the authors suggest might be explained by the gastrointestinal digestion of ingested proteins. However, this would require that other bioactive components of the leaf also exert hypoglycemic activity since oral administration is the preferred route and one that has shown significant effect on glycemia in many studies. The hypoglycemic effect of the intraperitoneally administered protein isolate was not accompanied by insulin increase, which suggests that the effect is not caused by stimulation of insulin secretion [53].

### 3.3. Experiments with M. oleifera Methanolic or Ethanolic Extracts

In the case of the experiments performed in animal models with organic solvent extracts, it is necessary to point out that leaves, seeds, pods and bark have been employed as raw material, which may greatly influence the amount of bioactive substances in those extracts. However, due to similarity in experimental model designs they are reported together.

#### 3.3.1. Acute Effects

Oral administration of MO ethanolic extract at 500 mg/kg altered the hyperglycaemic condition of fasted, STZ-induced Type-2 diabetic rats after 90 min compared to the control group [36]. This extract also showed a significant effect on glucose tolerance one and two hours post oral glucose challenge. At these time points lower blood glucose levels were found in treated animals. However, there was no significant increase in plasma insulin levels at different times during the 2 h following a single dose of an ethanolic extract from plants cultivated in Bangladesh [36]. These authors assayed potential methods implicated in in vivo and in vitro assays and concluded that glucose absorption was probably reduced as a consequence of reduced α-amylase activity which led to lower carbohydrate digestion, while insulin secretion was not involved.

#### 3.3.2. Long-Term Effects Measured through Glucose Oral Challenge

A hypoglycemic effect was observed in HFD-fed mice when comparing those treated with fermented MO leaves methanolic extract (250 mg/kg) for 8 weeks and untreated mice [54]. On the contrary, no differences were found when mice were treated with non-fermented MO extract. The fermentation was made with three *Lactobacillus* strains from cabbage kimchi and the treatment with the methanolic extract resulted in lower blood glucose levels at 60, 90 and 120 min during an OGTT performed at the end of the study. Another experiment in response to glucose challenge, intraperitoneal this time, resulted in lower glucose levels during the 2 h following glucose injection in alloxan-induced diabetic rats that had received a MO leaves methanolic extract (300 or 600 mg/kg) daily during 6 weeks by oral gavage (Nigeria cultivation) [55]. The results were similar to the effect of metformin. AUC for glucose was more than 50% reduced with the two doses of the methanolic extract and with metformin compared to the untreated diabetic rats. Alongside, the methanolic extract improved insulin release [55]. A different study, performed with MO seeds ethanolic extract (161 mg MO isothiocyanate [MIC-1]/kg) showed significant blood glucose reducing effects in OGTT performed at weeks 2nd, 4th, 6th, 9th and 12th of treatment in mice with very-high fat diet-induced diabetes compared with untreated diabetic animals [56].

Regarding postprandial insulin levels in plasma, the different results observed with the organic solvent extracts of MO leaves can be explained by the different experimental designs. Firstly, different chemicals were used to induce diabetes and at different rodent’s age, i.e., adult rats [55] compared to neonatal rats [36], respectively, and secondly, the MO extracts were given during very different periods of 6 weeks [55] and single dose [36] in each of the experiments. It cannot be ruled out that the ethanolic extract could induce insulin release with longer periods of treatment in STZ-induced diabetic rats [36]; however, and thirdly, the different origin of the MO plants, i.e., Nigeria vs. Bangladesh, and the extraction methods employed can also influence the underlying hypoglycemic mechanisms due to the different composition of the leaves and extracts.

#### 3.3.3. Long-Term Effects Measured through Fasting Blood Glucose

There is a substantial number of animal studies assessing the effect of MO extracts obtained with organic solvent extractions on blood glucose levels after days or weeks of treatment. The fasted conditions of the animals at blood withdrawal is the general rule in these studies, although in certain works this fact is not clearly stated. In this sense, lower fasting blood glucose levels were found in both STZ-induced diabetic rats and normal rats after administration of 250 mg/kg MO methanolic extract (Nigeria cultivar) daily for 6 weeks compared to their respective control rats [57]. Coincident results were found in a similar model in two different works with Indian MO; the first one an 8-wk intervention with the methanolic extract (300 mg/kg) [59] and the second one with a methanolic extract from pods instead of leaves (150 mg/kg or 300 mg/kg), which was orally administered for 3 weeks [60]. An increase in insulin was also noted in both of these experiments. Lower fasting blood glucose and increased insulin was also found in *db*/*db* mice after oral administration of an ethanolic extract of MO leaves from Cambodia (150 mg/kg) for 5 weeks (from 483 to 312 mg/dL) [62]. Fasting blood glucose was also lower in alloxan-induced diabetic rats that had received an ethanolic extract (200 mg/kg; Nigeria cultivation) twice daily for 5 days compared to those measured in the control group [63]. A different experiment confirmed this finding in the same animal model but treated with an ethanolic extract of MO stem bark daily (250 mg/kg, India cultivar) for one week [64]. Opposite to all these positive results, the study by Olurishe et al. [65] failed to find a long-term improvement in fasting blood glucose with animals receiving an ethanolic extract (300 mg/kg) of MO leaf (Nigeria cultivar), in this case during 6 weeks. Improvements were found until day 21st compared to the hyperglycemic levels measured on day 1st, both in the MO treated group and in the group receiving a combined treatment with MO extract and sitagliptin (an anti-diabetic agent); however, the beneficial effects were not maintained at 35th and 42nd days of treatment. On the contrary, end-of-study glucose levels were significantly lower than day 1st levels if rats were treated with sitagliptin alone, thus indicating that the combined treatment had a less antihyperglicemic effect, may be due to drug-herb interactions. In a similar experiment, a 24-day long intervention with ethanolic and methanolic extracts form MO leaves (Nigeria cultivar) performed in alloxan-induced diabetic rats, led to significant reductions of around 70–85% in fasting blood glucose compared to diabetic control rats [48].

The study by Raafat and Hdaib [66], in alloxan-induced diabetic mice given intraperitoneal injections of *n*-hexane extract of MO seeds (40, 60 or 80 mg/kg) for 8 consecutive days reported a 55%, 62% and 70% decrease in blood glucose, respectively, on the 8th day. Blood glucose was measured 6 h post injection every other day. A similar effect was also observed when mice were injected with a chromatographic fraction of this extract identified by GC-MS as β-sitosterol. Both, MO extract and β-sitosterol, led to significantly higher levels of insulin 8 weeks post-administration than in the diabetic control animals. This long-term insulin-secretagogue effect seems to be involved in glycemic homeostasis and in the lower levels of HbA1C also found in treated mice 8 weeks post-administration. Another study with a seed’s purified compound is that of Bao et al. [67]. A 95% pure Niazirin compound from concentrated aqueous extract of MO seeds was tested for its anti-hyperglycemic and anti-inflammatory effects in *db*/*db* mice being administered 10 or 20 mg/kg/day (oral gavage) for 4 weeks. This long-term study showed a significant decrease in fasting blood glucose compared to untreated animals and reduced insulin levels with improved HOMA-IR. Moreover, in an OGTT, significantly lower blood glucose was observed after 90 min and 120 min [67]. The lowering effect of Niazirin on blood glucose of diabetic rats had been previously observed by Wang et al., who have also documented similar effects with other phenolic glycoside compounds isolated from MO seeds and administered intravenously during 2 weeks [68].

The majority of long-term studies found positive results on fasting glucose and OGTT, however, the effect on insulin levels did not so such coincidence among studies. While some of them report an increase in fasting insulin levels others report a decrease [67] or no effect [36,49,53]. Although the studies are performed using different types of material, and different lengths of treatment, there does not seem to be a clear pattern leading to an explanation for the varying results. Lack of consistency is also found regarding the effect of MO methanolic or aqueous extracts on fasting blood glucose when administered to normal rats. Some studies report decreased values [58] while more frequently, works report no changes [40,43,49,59], although, once again, the intervention period has different durations.

## 4. Scientific Evidences of *M. oleifera* Effect on Glucose Control in Human Studies

Contrary to the abundant number of animal studies reviewed above, there is a paucity of published clinical trials in humans, and especially those which include an adequate number of subjects. In addition, relevant methodological data are missing in some of them. The eight human studies with glucose control measurements found in the literature are presented in Table 2. Two studies have been performed with administration of capsules/tablets of ground MO leaves to non-insulin dependent type 2 diabetic patients who received only oral anti-diabetic medications and dietary recommendations to reduce energy intake. In the first study, by Kumari et al. [72], 22 diabetic patients received MO treatment and nine did not, while in the second study, by Giridhari et al. [73], 60 patients were divided equally into two groups (MO treatment and control). The duration of the intervention was 40 and 90 days, respectively. In Kumari et al. study, daily intake of 8 MO g led to 26% decrease of postprandial glucose at the end of the intervention [72]. Meanwhile, in Giridhari et al. study, two tablets per day of an unknown amount of MO powder, reduced postprandial blood glucose level from 210 mg/dL to 191, 174 and 150 mg/dL, respectively, after the first, second and third month of supplementation (29% decrease) [73]. In this last study, HbA1C was also reduced after treatment (from 7.81 ± 0.51% to 7.40 ± 0.63%) while this was not the case in the control group. However, basal differences were observed between MO and control groups both in HbA1C and initial postprandial glucose, which decreases somehow the resulting evidence grade since the subjects under MO treatment started with a worse glucose control than the other group.

In another study, 60 postmenopausal, but otherwise healthy women, divided into two parallel groups, received, either, no supplement or 7 grams of MO leaves powder, respectively, during 3 months [71]. In this period, 13.5% decrease in fasting glucose was observed. Moreover, the number of women normalizing glucose values (i.e., <110 mg/dL) in the MO group was higher than in the control group. Some of these studies also report significant reductions in serum cholesterol and tryacylglyceride levels [72] and improvements in the level of antioxidant vitamins (retinol and ascorbic acid) and antioxidant exzymes (superoxide dismutase [SOD], glutathione peroxidase [GSH-Px]) and oxidative stress biomarkers (malondialdehyde [MDA]) [71].

A randomized-placebo controlled study was performed by Taweerutchana et al. [70], including 32 therapy-naïve DM patients whose glucose control was evaluated while undergoing a short-term therapy (28 days) with eight daily capsules of either MO (4 g/d) or placebo. The volunteers performed 9-point glucose measurements weekly with a glucometer, which included premeal and postmeal measurements. The results showed no difference in fasting plasma glucose, HbA1C, mean daily plasma glucose, mean premeal, and mean postmeal plasma glucose between MO leaf and placebo group. A non-significant reduction of 0.2–0.3% was found in HbA1C as compared to baseline in both treatment arms, which reflected that self-monitoring plasma glucose provides feedback that improves glycemic control through lifestyle changes. The same dose (4 g) of a similar manufactured product of MO leaf had previously shown in 10 healthy volunteers an increase in mean insulin secretion compared to placebo [74]. Insulin was measured at fixed intervals during six hours after MO ingestion, while otherwise fasting condition was maintained. The mean plasma insulin in MO and placebo groups were 4.1 ± 7.1 and 2.3 ± 0.9 µU/mL, respectively and the AUC of the insulin/glucose ratio was 74% higher in the MO group. However, fasting blood glucose was similar in both groups and always within the normal range. Since a similarity has been reported between proteins isolated from MO leaf and insulin [53], the increase in insulin secretion might be explained by cross-reactivity of the antibodies used in the immunoassay with these MO proteins or the peptides resulting from their gastrointestinal digestion. In addition, the lack of effect on fasting plasma glucose might be explained by the fact that the subjects were healthy and fasted, and the homeostatic mechanisms are expected to effectively prevent hypoglycemic effects in them. On the other hand, considering the more physiological 9-point study in DM patients [70], the discrepancy with the observations in animal models showing a glucose tolerance improving effect might be attributed to too small amounts of some of the bioactive compounds, such as moringinine or chlorogenic acid [70]. Thus, among the three studies performed with diabetic patients taking MO for 28, 40 and 90 days, respectively [70,72,73], none of them has clearly shown differences in blood glucose between treatment and control groups. However, changes from baseline have been reported in some of them in different parameters such as fasting blood glucose [72] or postprandial glucose [72,73] or HbA1C [73]. These findings should be pondered by bearing in mind that both of these studies have methodological flaws, such as a low number of subjects [72] or apparent differences in basal values between treatment and control groups [73].

In this respect, a higher dose, in the form of 20 g of MO dried leaf power administered as part of a meal, resulted in lower postprandial glucose levels compared to a control meal in a group of Saharawi diabetic patients [12]. Also, the increase in glucose from baseline was lower at 90, 120 and 150 min after the beginning of the meal. In contrast, healthy adults did not show any difference in postprandial glucose [12]. These findings suggest that MO could be useful in improving the glycemic response in populations which have no access to drug therapies. Another single dose study, performed in healthy adults who were administered MO tea has revealed that the lower dose of 200 mL induced a higher reduction in glucose levels 30 min after glucose overload (22.8%) compared to the higher dose (400 mL, 17.9%) [45]. However, the final glucose levels at 150 min were similar between treatment groups and significantly lower compared to the control group. These findings suggest the potential anti-glycemic effect of MO in healthy adults and provide new insights about the different impact and mechanism of action of different doses of MO, suggesting that the lower dose had a more potent effect on intestinal glucose absorption while the higher dose had a higher effect on circulating glucose [45]. In addition, the intake of cookies containing MO leaf powder revealed a significant blood glucose reduction 30 and 45 min after ingestion compared to control cookies, as well as a decrease in hunger ratings [69]. Based on the studies described here, the evidence in humans on the potential usefulness of MO as an adjuvant therapy to control hyperglycemia is so far limited but promising, especially that coming from postprandial studies. There is, however, partial inconsistency and methodological weaknesses in some of the published studies.

## 5. Dose Comparison between Animal and Human Studies

Comparing the doses employed in animal and human studies could help devise future steps in research. When considering the amount in the three diabetic animal experiments published using MO dry leaf powder as test material (Section 3.1) is necessary to pay attention to the administration procedure. The first study [40], via oral cannula needle, administered 50 mg/day, which, using the guide for dose conversion between animal and human by Nair and Jacob [80] would equate to 4.3 g for an 80 kg human. The second experiment in rats [39] administered 200 mg/kg b.w., equivalent to 2.6 g/d for human. The third experiment [41] using 2 or 4% MO leaf powder mixed in the diet, would approximately amount to 2650 and 5300 mg/kg b.w., respectively, if 20 g of diet are consumed per day, which for a human would equate to 34 g/day and 68 g/day. This amount of dry leaves could be in the upper range of a safe intake since there is a recommendation not to exceed 70 g of fresh leaves per day in long term consumption [81]. However, the actual grams eaten per day by the rats in the experiment is not reported [41]. Dry leaves have a higher concentration of phytochemicals and nutrients (except vitamin C) than fresh leaves per 100 g of matter [4,82]. Although antinutrients might be an issue here, MO has been found to contain a relatively low amount of them (phytates, saponins, tannins and oxalates) [83]. Overall, it seems that doses used in animal studies that have shown hypoglycemic effects in diabetic rats are within the physiological range. However, most of the experiments in animals have been performed with dried (or freeze-dried) leaves extracts. In humans, with the exception of the published work of Fombang and Saa [45] in 15 healthy individuals who consumed a hot MO leaf powder tea (200 mL and 400 mL or control; 5 subjects per group) prior to a OGTT, no clinical trials have reported effects of aqueous or alcoholic preparations of MO leaves. A typical dose of an extract in rodent’s experiments is around 200–300 mg/kg b.w. and no adverse effect have been observed with a methanol extract up to a dose of 3000 mg/kg [84]. Thus, it seems possible to obtain positive results for glycemic control with physiological doses of the extracts in animal experiments. Translating this to humans would require well powered human studies to evaluate the dose-effect responses within human adjusted ranges as well as, simultaneously, providing evidence that no safety issues arise with these preparations. According to Fahey et al. [10]; a cold tea preparation (8 ounces or 227 mL) of MO chopped or powdered leaves contains little protein but enough glucomoringin (200 µmol or more; at least 25% of the hot water molar yield of glucomoringin) and also myrosinase activity to provide significant biological activity through conversion of the former to the isothiocyanate moringin. On the contrary, hot teas prevent this conversion because of the inactivation of the heat-sensitive enzyme. In this case, the bioavailability of isothiocyanates from glucosinolates would rely on the unique gut microbiome activity of each individual. Thus, performing clinical trials with cold water extracts of MO leaves has been encouraged [10] and should be exempt of safety issues at reasonable intake levels.

## 6. Scientific Evidences of *M. oleifera* Effect on Metabolic Alterations Related to Diabetes

Induction of diabetes is a convenient model to study not only glucose control but also other parameters that are altered as a result of diabetic complications, such as those resulting from increased hyperglycemia-induced oxidative stress. Thus, serum lipid profile, antioxidant enzymes and lipid peroxidation, inflammatory cytokines and microscopic lesion observations of different tissues and organs have been measured in many diabetes-induced animal studies. These parameters are associated with the neuropathy, nephropathy, retinopathy, cardiovascular and other pathophysiological manifestations in diabetic disease [42,51,57].

As shown in Table 1, and in more detail in Table 3 and Table 4, scientific works are accumulating in which an extended number of pathophysiological manifestations that frequently coexist with glucose intolerance or diabetes are studied. For the sake of systematization these are divided in two wide areas in this review, namely, (1) antioxidant capacity and inflammation (Table 3), and (2) lipid accumulation, histopathology of damaged organs and gene expression of proteins involved in metabolic pathways (Table 4). However, diabetes complications originate from a complex interrelated system of cellular and metabolic mechanisms which are not independent from one another. One of them is the increased production or ineffective scavenging of ROS [85]. One outcome of excessive levels of ROS is the modification of the structure and function of cellular proteins and lipids, leading to overall dysfunctional biological activity, immune activation and inflammation [86]. In this sense, numerous studies have consistently shown that different parts of MO, mainly leaves [41] and seeds [41,42], but also pods [60] have potent antioxidant capacity in animal models of diabetes. Indeed, these parts of MO are able to repair the high levels of oxidative stress that are intrinsic to the chemical induction of diabetes in the animal models and in animals fed a very high-fat diet. This effect has also been observed with plant extracts, both aqueous and obtained with organic solvents, and documented in many different organs, such as the liver [46,52,61], kidney [57], pancreas [50], brain [76] and heart [59]. The election method employed to test the antioxidant capacity in these organs is the measurement of lipid peroxidation through the formation of MDA, which is consistently diminished in diabetic animals treated with MO compared to untreated animals. This might be explained by the effect of MO on the activity of antioxidant enzymes, since many study results confirm that MO and its extracts can reverse the decrease in SOD, catalase (CAT) [42,46,52,59,60,76] and GSH-Px [41,59] observed in diabetic animals compared to the negative control, as well as increase the non-enzymatic antioxidant system GSH (reduced glutathion) [41,42,46,50,59,60,61], which is also decreased in the diabetic condition. The results, as shown in Table 3, are very consistent among studies and in all type of organs; only Paula et al. study [53], performed with a protein isolate of MO leaves, showed increase in CAT but no effect on SOD activity, perhaps due to a short treatment duration of only 7 days. On the other hand, controversy also seems to arise when normal control rats instead of diabetic rats are used to test the methanolic extract of MO. In this case, no significant differences were observed in the levels of SOD, CAT, GSH-Px and GSH although consistent trends pointed towards higher levels of these antioxidant molecules [59].

Regarding inflammation, positive effects of the oral administration of MO and its extracts have been observed when measuring the expression of inflammatory cytokines in the liver and muscle [54], kidney [62] and wound tissue [75] of diabetic animals. Similar results were found when measuring the concentration of inflammatory cytokines in liver [58], kidney [42,57], and plasma or serum [28,42,67]. Specifically, this reduction of inflammation has been observed on TNF-α, IL-6 [28,54,58,62,75] and iNOS [56,62,75] levels. In addition, lower concentrations of other inflammation markers such as IL-1β [28,62,75], MCP-1 [58] and COX-2 [62,75] were also found. On the opposite, an increase in the anti-inflammatory cytokine IL-10 [67] and also in IFN-γ production by peripheral blood mononuclear cells [49] have been described in MO supplemented diabetic mice.

Other extensive information derived from the same experimental models and interventions with fresh, dried or variably processed parts of the plant MO, evidences that it has an effect in the structural and functional protection of different organs that can be partially evaluated through histological analysis of tissue sections (Table 4). In this sense, Omodanisi et al. [58] observed that oral administration of a methanolic extract of MO (250 mg/kg) to STZ-induced diabetic animals for six weeks induced a significant reduction in altered hepatic enzyme markers compared to untreated diabetic rats and the hepatoprotective effect was also proven with the histopathology analysis of liver sections. Reduction of hepatic enzymes was also observed by Khan et al. [43], as well as the prevention of liver histopathological changes in a number of different studies [46,67]. Improvement in histopathological findings associated to diabetes induction have also been documented in renal tissue [62], pancreas [42,46] and heart [59]. Lower hepatic lipid accumulation has also been evaluated through microscope observation in response to fermented MO [54], and in homogenate quantification after administration of extracts from seeds [56], leaves [28] and niazirin [67], respectively.

Regarding the analysis of the serum lipid profile, the results seem to be consistent about the hypolipidemic effect of MO. A significant decrease of LDL-cholesterol and triacylglycerides are reported in different works comparing treated and untreated diabetic animals [43,55,62,67]. Some of them also find decreased total cholesterol levels [43,55,58] and increased HDL-cholesterol with MO extracts [43,55] and MO seeds-purified niazirin [67]. Khan et al. [43] observed a smaller level of restoration of lipid profile in STZ-induced diabetic mice than in HFD-induced diabetic mice.

Finally, efforts have been also made to shed light into the molecular mechanisms that lead to lower lipotoxicity and lipid uptake during treatment with MO in diabetic models. In this line, Joung et al. [54], measured the expression of genes involved in liver lipid metabolism and found a decreased expression of the lipogenic genes ACC, FAS, SREBP-1 with fermented MO leaf and increased lipolytic gene expression as indicated by CD36, ACOX-1, ATGL and HSL measurements. No effect, however, was observed on the expression of the lipogenic or lipogenesis regulatory genes C/EBPα, PPAR-γ, LPL and pAMPK and the lipolytic CPT1 and PPAR-α. They also found decreased endoplasmic reticulum stress in skeletal muscle as shown by lower expression levels of the protein chaperon BiP and the disulfide bond regulator PDI under MO treatment of the diabetic animals. Coincident findings regarding reduced SREBP-1 and FAS expression were found by other authors [67] who also reported an increase in fatty acid oxidation as documented by increased ratio p-ACC/ACC in the liver of *db*/*db* mice treated with niazirin from MO. Since lipotoxicity and insulin resistance are closely related as metabolic abnormalities present in diabetes development, the expression of genes involved in glucose homeostasis has also been studied. Abd Eldaim et al. [46] found that a MO aqueous extract increased the expression of glycogen synthase gene (glycogenic), decreased that of pyruvate carboxylase (gluconeogenic) and the apoptotic caspase 3 gene in liver, which prove the normalization exerted by MO treatment on the dysregulated metabolism associated to diabetes induction. This was further supported by findings showing that the treatment with niazirin inhibited the abnormally high activity of the gluconeogenic enzymes G6Pase and PEPCK in the liver of untreated *db*/*db* mice [67]. It also repaired glycolytic activity in the liver by increasing the hexokinase and pyruvate kinase enzymes which were decreased in *db*/*db* mice [67].

## 7. Final Remarks

The extent of investigation on the potential use of MO plant in the control of glycaemia is still scarce. The number of animal studies is reasonably high and covers an ample range of different study designs trying to shed light on the observed pathophysiological changes in models of induced hyperglycemia. A majority of the results presented show significant improvements in blood glucose, both in fasted state and in response to a glucose tolerance test. The mechanisms of action revealed from the animal model experiments presented here include the normalization of the gene expressions of enzymes involved in glucose metabolism resulting in the restoration of liver glycolytic activity and glycogen storage [46,55,67] as well as reducing gluconeogenesis [28,46,67] and improving insulin signaling [28]. Delayed gastric emptying can also improve glycemic control in the postprandial state [39], which could be related to the high fiber content in the MO leaves. In addition, inhibition of intestinal glucose uptake has been shown in several animal models [36,39,43] and has been related to the inhibition of glucose transporter proteins in cell membranes by flavonoid glycosides [39]. On the contrary, improved glucose uptake was found in muscle and liver [43,55], favoring reduction of blood glucose and insulin-like proteins isolated from the MO plant could also contribute to the hypoglycemic effect [53]. Regarding clinical trials in humans, there are only a few published studies and with very variable designs. Thus, it is difficult to reach consensus about the indication of MO as an adjuvant therapy in the prevention and treatment of DM. Thus far, postprandial studies seem to offer more evidences of the hypoglycemic effect of MO leaves than the daily dose interventions lasting 28 to 90 days. More intervention studies in diabetic or prediabetic patients are certainly needed with more stringent inclusion criteria and a sufficient number of patients. It is also highly recommended to try to quantify the bioactive substances administered with the experimental material tested to facilitate comparison between studies. In this line, efforts made to characterize the composition of different MO tree varieties and other crop influential factors will add positive knowledge to a more rational design and use of MO food supplements. On the other hand, the use of extraction methods to obtain preparations enriched in specific bioactive compounds requires further research before they can be implemented for the therapy of human disease, since safety during prolonged use should be warranted prior to this step. Nonetheless, tea preparations, especially with cold water, could provide enough glucosinolates and its isothiocyanate metabolites for a potential biological effect, provided moringin is, at least to some extent, responsible for the hypoglycemic effect of the plant. Finally, the composition of the extracts and how the different ingredients interact through additive, synergistic or inhibitory effects should be investigated because this will impact the therapeutic use of the extracted preparations or even the isolated compounds as compared to the raw materials.

## Figures and Tables

**Figure 1 nutrients-12-02050-f001:**
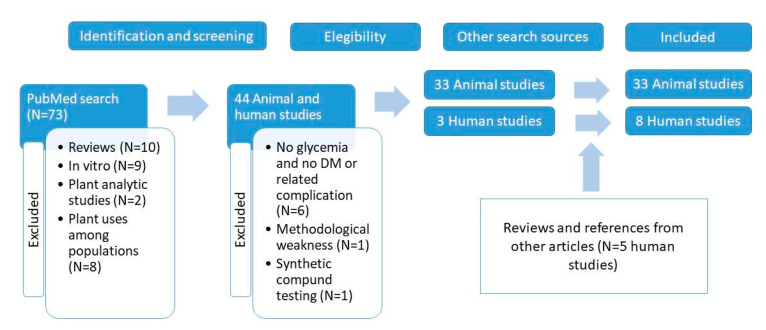
Flow chart of the selection of the animal and human studies included. DM: diabetes mellitus.

**Table 1 nutrients-12-02050-t001:** Animal studies on the effect of *M. oleifera* on glucose control and biomarkers related to diabetes complications.

Model	Treatment and Duration	Measurements	Evidences: MO Treated Animals vs. Untreated	Ref
**Studies with raw MO**				
Alloxan-induced diabetic Sprague Dawley rats. Normal Sprague Dawley rats	MO dry leaf powder (50 mg/day, gavage) 8 wk.	Body weight, BG, lipid profile. Intestine histopathology, lactic acid bacteria and Enterobacteriaceae (culture).	↓ BG. Prevented weight loss. No effect on lipid profile. No histopathology observations. No effect in BG in normal rats. ↓ Enterobacteria enumeration	[40]
STZ-diabetic male Wistar rats	Diets containing 2% and 4% MO leaves or MO seeds (± Acarbose, ACA) 2 wk.	BG (every 3 days), acetylcholinesterase (AChE), butyrylcholinesterase (BChE)] angiotensin-I converting enzyme (ACE), arginase, CAT, GST and GSH-Px activities, GSH and nitric oxide (NO) levels in brain	↓ BG (all treated groups). The highest reduction occurred with 4% MO leaves + ACA. ↓ AChE, BChE and ACE activities and ↑ antioxidant molecules (both preventive of cognitive dysfunction)	[41]
Male spontaneously diabetic Goto-Kakizaki rats Normal male Wistar rats	MO leaf powder (200 mg/kg) Single dose (glucose-MO solution, oral admin.)	OGTT (BG at 10, 20, 30, 45, 60, 90 and 120 min and iAUC). Stomach, small intestine and caecum content weights.	↓ BG at 20, 30, 45, and 60 min. ↓iAUC. ↑ stomach contents (⇒delayed gastric emptying). ↓ BG at 10, 30 and 45 min in normal rats.	[39]
STZ-induced diabetic adult male Wistar rats	MO seed powder (50 or 100 mg/kg) in the diet 4 wk.	FBG, HbA1c, lipid peroxidation, antioxidant enzymes, liver and renal function, IgG, IgA, serum and kidney IL-6 and kidney and pancreas histopathology	Prevented weight loss, ↓ FBG (35% and 45%, 50 and 100 dose, resp.) and ↓ HbA1C (13% and 22%). Improvement of all oxidative status parameters, Igs and IL6; all approaching the negative control. Restoration of the normal histology of both kidney and pancreas. Both doses are effective but, overall, the higher dose is more effective.	[42]
**Studies with MO aqueous extract**				
STZ-diabetic female Wistar rats HFD-induced diabetic C57BL/6 mice Normal rats and mice	1) Aq MO, 100 mg/kg (in the diet) to STZ-induced rats 2) Aq MO, 200 mg/kg (in the diet) to HFD-fed mice. Either, 2 single doses (2 days) or 3 wk.	FBG, OGTT, lipid profile, liver marker enzymes.	↓ FBG, ↓ 2 h glucose and AUC of glucose in OGTT. ↓ SGOT and SGPT (HFD- and STZ-induced animals), improved lipid profile (more significant in HFD-mice than STZ-diabetic rats) No differences in FBG and biochemical parameters in normal rats and mice	[43]
STZ-induced diabetic male Wistar rats (sub, mild and severely diabetic) Normal Wistar rats	Aq MO (100, 200 and 300 mg/kg, oral gavage) Either, single dose or 21 days	BG and OGTT in response to single doses. FBG, PPG, haemoglobin, total protein and weight gain after 21 days. BG and OGTT in response to single doses	↓ BG in OGTT (at 3 h post oral glucose): maximum fall of 31.1% and 32.8% in sub-diabetic and mild-diabetic rats, respectively, occurred always with the 200 mg/kg dose. ↓ FBG (69.2%) and ↓ PPG (51.2%) in severely diabetic rats on a 21-day treatment. ↑ Hemoglobin (10.9%) and total protein (11.3%) ↓ BG in normal rats: maximum fall of 26.7% occurred 6 h after 200 mg/kg, single dose. ↓ BG in OGTT (at 3 h post oral glucose) in normal rats: maximum fall of 29.9% with 200 mg/kg dose	[44]
Adult male Wistar rats	MO tea (10, 20 and 30 mL/kg, gavage) Single dose	OGTT (4 g/kg b.w. of glucose, 30 min after MO tea)	↓16, 18 and 6% total PPG ⇒ Lower doses are more efficient.	[45]
Alloxan-induced diabetic Wistar rats. Normal Wistar rats	Aq MO (250 mg/kg, oral admin.) 18 days	BG, hepatic lipid peroxidation and antioxidant enzyme activities, histoarchitecture of hepatic and pancreatic tissues, gene expression of glycogen synthase (GS), pyruvate carboxylase (PC) and caspase 3, and SOD and CAT activities.	↓ BG, prevented organ changes and significantly restored all measures. Normalized the expression of apoptotic, gluconeogenic, and glycogenic genes in hepatic tissue No effect on BG in normal rats. ↑ liver GSH and GS expression. No other significant changes in normal rats.	[46]
Alloxan-induced diabetic female Wistar rats. Normal Wistar rats	Aq MO (250 mg/kg, oral admin.) 18 days	Body weight, BG, lipid profile, lipid peroxidation, histoarchitecture of hepatic and pancreatic tissues, expression of pyruvate kinase (PK), pyruvate carboxylase (PC), and fatty acid synthase (FAS) in liver.	↓ BG ↑ Body weight, ↓ TG and MDA. Normalized the expression of enzymes of gluconeogenesis and fatty acid synthesis. Normalized histological structures. No effects in normal rats.	[47]
Alloxan-induced diabetic Wistar rats	MO extracts: Aq, Me MO and Et MO (50% water: 50% alcohol and 100% alcohol, dose range: 200–400 mg/kg, oral admin.) 24 days	FBG	All extracts and doses ↓FBG around 70–87%. Aq MO (300 mg/kg) reduction was 82%. All extracts showed body weight restoration capacity.	[48]
Alloxan-Induced diabetic albino mice Normal albino mice	Aq MO (100 mg/kg, oral gavage) 14 days	FBG, insulin, HOMA-IR, TAC, creatinine, blood urea nitrogen (BUN). Percentage CD44, CD69 and IFN-γ positive cells in PBMCs	↓FBG, ↑(NS) insulin, ↓ HOMA-IR. ↓creatinine and BUN and ↑ TAC and IFN-γ. ↑ Insulin in normal rats and no effect on FBG and HOMA-IR. No effect in other parameters measured.	[49]
STZ-induced diabetic male Sprague-Dawley rats Normal Sprague-Dawley rats	Aq MO (200 mg/kg, oral gavage) 8 wk	FPG; GSH, lipid peroxidation, histopathology and morphometric analyses of pancreas	↓ FPG (62%) ↓ MDA and ↑ GSH, normalization of histopathological and morphometric changes No effects in normal rats.	[50]
STZ-induced diabetic rats	Aq MO (100 mg/kg, oral gavage) 24 wk.	Body weight, BG and HbA1C. TNF-α, IL-1β, VEGF, PKC-β, GSH, SOD, CAT in retinae. Retinal leakage and retinal vessel caliber (arteriolar and venular) and basement membrane thickness.	↓ BG (33%), and HbA1C (40%). Preserved weight gain. ↑Antioxidant parameters, ↓ inflammatory and angiogenic parameters, ↓ all morphological and structural alterations of the retinae.	[51]
VHFD-induced obese male C57BL/6J mice	Aq MO (5% MO in VHFD, 66 mg/d MIC) 12 wk.	OGTT at 4th, 8th and 12th wk., plasma insulin, leptin, resistin, IL-1β and TNFα, total cholesterol and triglycerides. Liver histology and gene expression: TNF-α, IL-1β, IL-6, G6Pase, PEPCK and GcK. Insulin signaling proteins (liver and muscle) and lipolysis-related gene expression and protein levels (adipose tissue and liver).	↓BG and ↓ AUC of glucose at 8th and 12th wk (NS ↓ at wk. 4th). ↓ Weight gain, ↓ body fat accumulation, ↓ plasma insulin, leptin, resistin, cytokines and cholesterol, ↓ Hepatic G6Pase and TNF-α expression, improved insulin signaling (↑IRSs, protein kinases, PI3K, and GLUT4) and liver lipolytic protein levels.	[28]
High fructose diet-induced diabetic male Sprague Dawley rats	Aq MO (300 mg/kg, oral admin.) 4 wk.	FSG, Insulin, HOMA-IR, testosterone and FSH; MDA, SOD, CAT in liver; insulin receptor (IR), IRS-1, GLUT-4 & GLUT-5 and SOD, steroidogenic acute regulatory protein (StAR) and 3β-hydroxysteroid dehydrogenase (3β-HSD) expression in liver	No effect on FSG; ↑ insulin and ↓ HOMA-IR, ↓ MDA, ↑ SOD, CAT and testosterone. Improvement of the down-regulation of the insulin signaling pathway. Improved regulatory proteins of testicular function	[52]
Alloxan-Induced diabetic Unib:SW (Swiss) mice	Protein isolate of MO leaves (500 mg/kg) (i.p. and oral admin.) Single dose and 7 days	BG (5 h after single dose and 4 h after daily treatment at 3rd and 7th day), insulin (5 h after single dose), liver MAD, CAT and SOD.	i.p. admin.: ↓BG, ↓ liver MDA and ↑CAT. Did not change serum insulin. Oral admin.: No effect on BG (due to protein digestion).	[53]
**Studies with MO methanolic or ethanolic extracts**				
STZ-induced diabetic Long Evan rats	Et MO (0.5 g/kg, oral admin.) and Et MO with glucose (OGTT). Single dose, both	FBG, OGTT (2.5 g/kg oral glucose), insulin, intestinal glucose absorption by perfusion technique through the pylorus.	↓ FBG at 90 min post Et MO and ↓ BG in OGTT. No change in insulin. ↓ Glucose absorption.	[36]
HFD-induced obese male C57BL/6J mice	Me MO (250 mg/kg, oral admin.), fermented (FM) and non-fermented (NFM). Fermentation starter: 3 LAB strains isolated from cabbage kimchi 10 wk.	Glucose tolerance (2 g/kg glucose, ip. injection) at 8th wk., hepatic lipid accumulation, expression of proteins and genes involved in glucose and lipid regulation	FM: ↓ AUC glucose, ↓ hepatic lipid accumulation. Upregulation of genes related to lipid metabolism. ↓ Oxidative stress and lipotoxicity in muscle. ↓ proinflammatory cytokine expression in muscle, and liver tissues. NFM: No effect on glycemic response. ↓ Hepatic lipid accumulation. Mixed effects on proinflammatory cytokine expression and levels in different tissues.	[54]
Alloxan-induced diabetic rats	Me MO (300 or 600 mg/kg, oral gavage) 6 wk.	Food intake, body weight, intraperitoneal glucose tolerance (IPGT, 30, 60 and 120 min post 2 g/kg glucose administration), serum glucose, insulin, and lipids, liver and muscle glycogen synthase activity, glycogen content, and glucose uptake.	Prevented weight loss, ↓BG and ↑Insulin, ↓BG at all-time points in IPGT. Improved lipid profile, increased glycogen synthase activity and glycogen content in muscle and liver and improved glucose uptake.	[55]
Obese C57Bl/6J male mice fed VHFD Normal C57Bl/6J male mice fed LFD	Et MO (seed, 47% MIC) in the diet. Average dose: 161 ± 19 mg MIC-1/kg in VHFD; 335 ± 23 mg MIC-1/kg in LFD. 12 wk.	Body weight, body composition, OGTT at week 2nd, 4th, 6th, 9th and 12th; liver lipids, IL-1β, IL-6, TNF-α, iNOS, NQO-1 gene expression, intestinal microbiota composition and load.	↓ AUC glucose in the VHFD animals. Similar AUC glucose in LFD animals (except at week 9, treated < untreated). ↓ body weight, ↓ adiposity; ↓ iNOS expression, ↑ antioxidant NQO-1 expression (both diets); ↓ bacterial load, modulation of bacterial community (both diets)	[56]
STZ/HFD-induced diabetic male Wistar rats Normal Wistar rats	Me MO (250 mg/kg, gavage) 6 wk	FPG, kidney lipid peroxidation, CAT, GPx, SOD activities, GSH and inflammatory biomarkers.	↓ FPG, ↓ kidney weight and relative kidney weight. ↓ Kidney MDA, ↑ CAT (and ↑ NS SOD), ↓ pathological observation in histology. ↓ FPG, ↑ SOD, ↓ GPx (and ↑ NS CAT) in normal rats.	[57]
STZ-induced diabetic female Wistar rats Normal Wistar rats	Me MO (250 mg/kg, gavage) 6 wk.	FSG, liver weight and enzymes, lipid profile, liver antioxidant capacity, inflammatory cytokine levels and histopathology.	↓ FSG ↓ Liver weight, SGOT, ALP, LDL and cholesterol. ↓ IL-6, TNF-α, and MCP-1. Improvement of liver histological alterations Non-significant ↓ FSG in normal rats.↓ Liver enzymes and ↑ HDL in normal rats.	[58]
STZ-induced diabetic male Sprague Dawley rats. Normal male Sprague Dawley rats.	Me MO (300 mg/kg, oral gavage) 60 days	FBG (glucometer), FSG (ion exchange), HbA1C, Insulin, SOD, CAT, GPx, GR, GSH, TBARS.	↓ FBG, FSG, and HbA1c and ↑ increased insulin. ↑Activities of antioxidant enzymes in the heart of diabetic rats. No effects in normal rats.	[59]
STZ-induced diabetic Wistar rats Normal Wistar rats	Me MO (pods, 150 and 300 mg/kg, oral admin.) 3 wk.	BG, insulin, total protein and albumin and NO. Pancreatic lipid peroxidation, SOD, GSH, CAT and glycogen and histopathology.	↓ BG, ↑ insulin, total protein, albumin and NO. ↓ MDA and ↑ antioxidant activities in pancreas and reversed the histoarchitectural damage. No changes in normal rats.	[60]
STZ-diabetic Wistar rats	Me MO (200 mg/kg, gavage) 3 wk.	FBG, weight, supercomplex formation, ATPase activity, ROS production, GSH and GR levels, lipid peroxidation and protein carbonylation of liver mitochondria	↓ FBG: 86 ± 4.2 mg/dL vs. 229 ± 9.05 mg/dL. ↑ GSH and GR; ↓ lipid peroxidation and protein carbonylation of liver mitochondria.	[61]
*db*/*db* mice	Et MO (150 mg/kg) 5 wk.	FPG, lipid profile, kidney histology and expression of inflammation markers.	↓ FPG (36%), TG and LDL. ↑ insulin. ↓ kidney histopathological damage. ↓ expression of kidney inflammatory markers	[62]
Alloxan-Induced diabetic Wistar rats	Et MO (200 mg/kg, twice daily, oral admin.) 5–6 days	FBG, electrolytes: potassium (K), sodium (Na), chloride (Cl−), bicarbonate and lactate dehydrogenase (LDH) enzyme.	↓FBG. No effect on weight. No cytotoxicity. ↓ Bicarbonate, ↑ anion gap value (⇒acidosis, but lower that metformin).	[63]
Alloxan-induced diabetic Charles Foster strain male albino rats	Et MO (stem bark, 250 mg/kg, oral admin.) 1 wk.	FBG and urine sugar	↓ FBG and nil urine sugar detected.	[64]
Alloxan-Induced diabetic rats with/w-out sitagliptin treatment	Et MO (300 mg/kg, oral gavage) 42 days	FBG and other glycaemic control parameters, insulin, body weight, retinal microvasculature on lenticular opacity/morphology	No difference in FBG between day 42nd and day 1st. ↓ random BG (42nd vs. 1st day). Overall, less anti-hyperglicemic effect than sitagliptin alone. No changes in insulin secretion and body weight. No prevention or amelioration of retina lesions.	[65]
Alloxan-induced Swiss-Webster male mice	A) N-Hexane extract of MO seeds (40, 60 and 80 mg/kg, i.p.) B) β-sitosterol (BSL) fraction from hexane extract (18, 25 and 35 mg/kg, i.p.) Single dose (acute) and 8 days	BG (acute: 1, 2 and 6h; subchronic: 1st, 3rd, 5th, and 8th day, 6 h post daily injection); Insulin, HbA1C, CAT, lipid peroxidation (8 wk post-treatment end); Diabetic painful neuropathy measures (hot plate latency, tail flick latency and von Frey filaments test, 8 wk post-treatment end)	Acute: ↓ BG at 2 h and 6 h with all doses of MO and BSL. Subchronic: ↓ BG from day 1 to day 8 with MO and BSL (maximum ~ 70% reduction on day 8th). ↑ Insulin several fold, ↓ HbA1C, improved antioxidant markers, 8 wk. post-treatment end. ↑ body weight (both). Significantly improved thermal hyperalgesia and tactile allodynia (MO more powerful than BSL).	[66]
db/db mice	Niazirin (10 mg/kg and 20 mg/kg, extracted and concentrated from seeds. 95% purity. Oral gavage) 4 wk.	FBG and Insulin, HOMA-IR, OGTT. Plasma TNF-alpha, IL-10, LDL, HDL, TC, TG, NEFA levels; Liver glycogen, HK, PK, G6Pase and PEPCK activities; Liver histological analysis; Liver AMPK, p-AMPK, FAS, p-ACC, SREBP-1, PPAR-α, SirT1, FOXO1, HNF-4α, PGC-1α, PFKFB-3.	↓ FBG and insulin with improved HOMA-IR (both doses). OGTT: lower BG at 90 and 120 min (both doses) Improved glucose uptake and glycogen storage in the liver through AMPK pathway. Improved lipid profile; reduced FA synthesis and induced FA oxidation through AMPK pathway.	[67]
STZ-induced diabetic ICR mice	4 compounds isolated from MO seeds by macroporous resin adsorption and chromatography (20 mg/kg, i.v.) 2 wk.	BG	↓ BG, 3 compounds: 1) N,N′-bis{4-[(α-l-rhamnosyloxy)benzyl]}thiourea, 2) niazirin A, 3) S-Methyl-N-{4-[(α-l-rhamnosyloxy)benzyl]}thiocar bamate No effect: 4-[(6-deoxy-α-l-mannopyranosyl)oxy]-benzaldehyde	[68]

Extracts are from leaves unless otherwise specified; MO: *M. oleifera*; Et MO: ethanolic extract of MO; Aq MO: aqueous extract of MO; Me MO: methanolic extract of MO. BG: Blood glucose; STZ: Streptozotocin, OGTT: Oral glucose tolerance test; iAUC: Incremental area under the curve; CAT: catalase, GST: Glutathione S-transferase; GSH-Px: Glutathione peroxidase; GSH: Reduced glutathione; FBG: Fasting blood glucose FBG; HbA1C: glycated hemoglobin; Igs: Immunoglobulins; HFD: high-fat diet; SGOT: Serum Glutamic Oxaloacetic Transaminase; SGPT: Serum Glutamate Pyruvate Transaminase; PPG: Postprandial glucose; SOD: Superoxide dismutase; MDA: Malondialdehyde; HOMA-IR: Homeostatic model assessment for insulin resistance; TAC: Total antioxidant capacity; FPG: Fasting plasma glucose; VEGF: vascular endothelial growth factor; PKC: protein kinase C; FSG: Fasting serum glucose; VHFD: Very high-fat diet; MIC: *Moringa* isothiocyanates; G6Pase: Glucose-6-phosphatase; PEPCK: phosphoenolpyruvate carboxykinase; GcK: Glucokinase; AUC: Area under the curve; NS: Non-significant; IRSs: Insulin receptor substrates; PI3K: fosfatidilinositol- 3-kinasa; GLUT-4 and -5: Glucose transporter 4 and 5; i.p.: intraperitoneal; LFD: Low-fat diet; TNF-α: Tumor necrosis factor α; iNOS: Inducible nitric oxide synthase; NQO-1: NAD(P)H dehydrogenase [quinone]-1; FSG: Fasting serum glucose; ALP: Alkaline phosphatase; LDL: Low density lipoprotein; MCP-1: Monocyte chemotactic protein 1; GR: Glutathion reductase; NO: Nitric oxide; TG: Triacylglycerides; HDL: High density lipoprotein cholesterol; NEFA: Non-esterified fatty acids; HK: Hexokinase; PK: Piruvate kinase; AMPK: 5′-AMP activated protein kinase; p-AMPK: Phosphorylated AMPK, FAS: Fatty acid synthase; pACC: Phospho acetyl-CoA carboxylase; SREBP-1: Sterol regulatory element-binding protein; PPAR-α: peroxisome proliferator-activated receptor; SirT1: Sirtuin 1; FOXO1: Forkhead box protein O1; HNF-4α: Hepatocyte nuclear factor 4 alpha; PGC-1α: Peroxisome proliferator activated receptor-1α; PFKFB-3: Phosphofructo-2-kinase/fructose-2,6-bisphosphatase-3; i.v.: Intravenous.↑:Higher in MO treated animals than control,↓: Lower in MO treated animals than control.

**Table 2 nutrients-12-02050-t002:** Evidence from human studies on the effects of *M. oleifera* on glycemic control and related parameters in healthy and type 2 diabetic adults.

Treatment and Duration	Study Design	Subjects	Measurements	ResultsCompared to Baseline	ResultsCompared to Control Group	Ref.
Meal containing MO leaf powder (20 g) C: Control meal. 2 single occasions	Randomized, placebo controlled, crossover, PP	10 healthy adults (6 W/4M)Age: 42 ± 11 y BMI: No data 17 type 2 diabetic patients (9 W/8 M) Age: 62 ± 9 y BMI: 25.2 ± 4.3 kg/m^2^	Fasting glucose (finger prick and glucometer) and postprandial glucose at 30, 60, 90, 120, 150, and 180 min from the beginning of the meal	--	Healthy: ↔ glycemic response Diab: ↓ blood glucose from 60 to 180 min. ↓ increment form baseline at 90, 120 and 150 min.	[12]
Cookies containing MO leaf powder (5% w/w) C: Control cookies Isocaloric and containing 50 g available carbohydrates 2 single occasions	Randomized single-blinded, placebo controlled crossover, PP	20 Healthy subjects (10 W/10 M) Age: 24.1 ± 1.33 y BMI: 22.0 ± 3.88 kg/m^2^	Fasting and postprandial blood glucose (finger prick and glucometer) at 15, 30, 45, 60, 90 and 120 min. Appetite, hunger and palatability scales	--	↓ non-significantly iAUC of glucose (P = 0.077). ↓ blood glucose at 30 and 45 min ↓ Hunger ratings	[69]
MO leaf capsules (4 g/d) C: Placebo capsules. 4 weeks	Randomized, placebo controlled, parallel	32 Therapy-naïve type 2 diabetics (15 W/17 M) Age: 50–60 y BMI: 27.5 kg/m^2^	9-point blood glucose (finger prick and glucometer) along 3 consecutive days.FPG and HBA1C levels.Creatinine and liver enzymes (ALT, AST)	↔ Fasting plasma glucose and HbA1C ↔ Creatinine, ALT, AST.	↔ Fasting plasma glucose and HbA1C ↔ mean daily BG, mean premeal, and mean postmeal BG. ↔ Creatinine, ALT and AST	[70]
MO leaf powder (7 g/d) in recipes in daily diet C: No supplementation 3 months	Randomized, controlled, parallel	60 Healthy postmenopausal women (60 W/0 M) Age: 45–55 y BMI: No data	FBG, hemoglobin, ascorbic acid, retinol, glutathione peroxidase, superoxide dismutase and malondialdehyde	↓ FBG↑ Blood haemoglobin ↑ Ascorbic acid, retinol, glutathione peroxidase and superoxide dismutase ↓ Malondialdehyde	↓ FBG, ↑ Blood haemoglobin ↑ Ascorbic acid and superoxide dismutase ↓ Malondialdehyde	[71]
MO leaf tea (200 mL or 400 mL) C: Distilled water Single occasions	Randomized, controlled, parallel, PP	15 Healthy subjects (0 W/15 M) Age: 20–29 y BMI:21.6 kg/m^2^ In 3 groups, N = 5	OGTT (50 g glucose) 30 min after MO tea oral dose (finger prick and glucometer)	--	↓ glycemia (17% [200 mL] and 19% [400 mL] reduction). Higher reduction at 30 min with lowest amount (22.8 vs. 17.9%)	[45]
MO leaf powder (8 g/d) C: No supplementation 40 days	Randomized, controlled	22 type 2 diabetics (8W/14 M) Age: 40–60 y BMI:18.5–35 kg/m^2^	FBG, PPG, lipid profile (methods unspecified)	↓FBG and PPG ↓ Total cholesterol, LDL and TAG	No statistical test performed	[72]
MO leaf tablet (2 units/day) C: No supplementation 90 days	Intervention controlled	60 type 2 diabetics on sulfonylurea medication (gender unspecified) Age: 40–58 y BMI: 20–25 kg/m^2^	HbA1c and PPG two hours after a meal	↓HbA1 (7.4% reduction) ↓PPG	↔ HbA1 and PPG	[73]
MO leaf powder capsules. Dosages: 0, 1, 2 and 4 g. C: 4 empty capsules. 4 single days separated by 2 wk.	Oral single dose study	10 Healthy volunteers (5W/5M) Age: 29 ± 5 y BMI: 18.5–23 kg/m^2^	Plasma glucose and insulin at intervals during 6 h after single dose of MO. Blood urea nitrogen, creatinine, AST and ALT at the first and forth visit	--	4 g: ↑ plasma insulin ↑ insulin AUC and ↑74% AUC of insulin/glucose ratio ↔ Plasma glucose, blood urea nitrogen, creatinine, AST and ALT	[74]

--: No applicable; ↔: No changes; ↓: Decreased; ↑: Increased; ALT: Alanine transaminase; AST: Aspartate transaminase; BG: Blood glucose; BMI: Body Mass Index; FBG: Fasting blood glucose; FPG: Fasting plasma glucose; HbA1c: Glycated haemoglobin; iAUC: Incremental area under the curve; LDL: Low-density lipoprotein; M: Men; MO: *M. oleifera*; PPG: Postprandial glucose; TAG: Triacylglicerides; W: Women.

**Table 3 nutrients-12-02050-t003:** Effects of *M. oleifera* on antioxidant capacity and inflammation protection in diabetic animal models.

MO Material	Animal Model	Organ or Biological Sample	Evidences MO Treated vs. Untreated	Ref.
**Gene expression of inflammation markers**				
Me MO (FM) ME MO (NFM)	Obese (HFD)	Liver and muscle	↓IL-6 and TNF-α. ↓IL-1β, only muscle ↓IL-6. ↓ (NS) TNF-α	[54]
Et MO	Diabetic	Kidney	↓TNF-α, IL-1β, IL-6, COX-2 and iNOS	[62]
Et MO (seed)	Obese (VHFD) Normal (LFD)	Liver and intestine	↓iNOS in intestine of VHFD and LFD. ↓iNOS in liver of VHFD. No effect on IL-1β, IL-6 and TNF-α. ↑NQO-1	[56]
Aq MO	Diabetic	skin wound tissues	↓TNF-α, IL-1β, IL-6, COX-2 and iNOS ↑ VEGF.	[75]
Aq MO	Obese (VHFD)	Ileum and liver Adipose tissue	↓TNF-α. ↓ (NS) IL-6 and IL-1β. ↑ Adiponectin	[28]
**Inflammatory cytokines**				
Me MO	Diabetic	Liver	↓TNF- α, IL-6 and MCP-1	[58]
Aq MO	Obese (VHFD)	Plasma	↓TNF-α and IL-1β	[28]
MOP (seeds)	Diabetic	Blood and kidney	↓IL-6	[42]
Aq MO	Diabetic and normal	PBMC	↑IFN-γ	[49]
Niazirin (seed)	Diabetic	Plasma	↓TNF- α ↑ IL-10	[67]
Oxidative status				
Me MO	Diabetic Normal	Liver	↑(NS) ORAC ↑ORAC	[58]
Me MO	Diabetic	Kidney	↓MDA	[57]
Me MO (pods)	Diabetic	Pancreas	↓MDA	[60]
Me MO	Diabetic	Liver (mitochondria)	↓MDA and ↓protein carbonilation	[61]
Aq MO	Diabetic	Serum	↓MDA	[47]
Aq MO	Diabetic	Liver	↓MDA	[46]
MOP (seed)	Diabetic	Serum and kidney	↓MDA	[42]
MO leaf/ MO seed	Diabetic	Brain	↓MDA	[41]
Aq MO	Diabetic	Brain, liver, kidney, pancreas and spleen	↓MDA	[76]
Aq MO	Diabetic	Pancreas	↓MDA	[50]
Aq MO	Diabetic	liver	↓MDA	[52]
MO (protein isolate)	Diabetic	Liver	↓MDA	[53]
Me MO	Diabetic	Heart	↓MDA, HP and CD	[59]
**Antioxidant enzyme activity**				
Me MO	Diabetic and Normal	Kidney	↑CAT (NS in normal) ↑SOD (NS in diabetic) ↓G-Px (NS in diabetic). No effect on GST	[57]
Me MO (pods)	Diabetic	Pancreas	↑GSH, SOD and CAT	[60]
Aq MO	Diabetic	Pancreas	↑GSH	[50]
Aq MO	Diabetic	liver	↑SOD, CAT	[52]
Aq MO	Diabetic	Liver	↑SOD, CAT, GSH	[46]
Aq MO	Diabetic	Plasma	↑TAC	[49]
MOP (seed)	Diabetic	Serum and kidney	↑CAT, SOD and GSH	[42]
MO leaf/ MO seed	Diabetic	Brain	↑CAT, G-Px, GST, GSH	[41]
Aq MO	Diabetic	Brain, liver, kidney, pancreas and spleen. Liver and pancreas	↑CAT, SOD ↑GST	[76]
MO (protein isolate)	Diabetic	Liver	↑CAT No effect on SOD	[53]
Me MO	Diabetic	Heart	↑CAT, SOD, G-Px, GR and GSH	[59]
Me MO	Diabetic	Liver (mitochondria)	↑GSH and GR	[61]
**Cholinergic dysfunction (associated to cognitive impairment as in diabetes encephalopathy)**				
MO leaf/MO seed	Diabetic	Brain	↓AChE, BChE and ACE	[41]

Extracts are from leaves unless specified in brackets; MO: *M. oleifera*; Me MO: Methanolic extract of MO; FM: Fermented MO leaves; NFM: Non-fermented MO leaves; Et MO: Ethanolic extract of MO; Aq MO: Aqueous extract of MO; MICs: MIC-1 (4-[(α-Lrhamnosyloxy)benzyl]isothiocyanate) and MIC-4 (4-[(4-O-acetyl-α-Lrhamnosyloxy) benzyl]isothiocyanate); MOP: MO powder; HFD: High-fat diet; VHFD: Very-high-fat diet; NS: Non-significant trend; IL-6: Interleukin-6; TNF-α: Tumor necrosis factor-α; IL-1β: Interleukin-1β; COX-1: Cyclooxygenase-2; iNOS: inducible nitric oxide synthase; NQO-1: NAD(P)H dehydrogenase [quinone]-1; TAC: Total antioxidant capacity; IFN-γ: interferon-γ; VEGF: vascular endothelial growth factor; MCP-1: monocyte chemotactic protein; IL-10: Interleukin-10; ORAC: Oxygen radical absorbance capacity; MDA: Malondialdehyde; TBARS: Thiobarbituric acid-reactive substances; HP: Hydroperoxides; CD: Conjugated dienes; CAT: Catalase; SOD: Superoxide dismutase; GSH: Glutathione; G-Px: Glutathione peroxidase; GST: Glutathione-S-transferase; GR: Glutathione reductase; AChE: Acetylcholinesterase, BChE: Butyrylcholinesterase; ACE: Angiotensin-I converting enzyme. ↑: Higher; ↓: Lower.

**Table 4 nutrients-12-02050-t004:** Effects of *M. oleifera* on lipids, histopathology and gene expression in diabetic animal models.

MO Material	Animal Model	Organ or Biological Sample	Evidences MO Treated vs. Untreated	Ref.
**Accumulation of lipids in tissues**				
Me MO (FM and NFM)	Obese (HFD)	Liver	↓ Hepatic adiposity (H&E staining)	[54]
Et MO (seed)	Obese (VHFD)	Liver	↓ Liver lipids (Folch’s method with modifications)	[56]
Aq MO	Obese (VHFD)	Liver	↓ Liver lipids (H&E staining)	[28]
Niazirin (seed)	*Diabetic*	Liver	↓ Hepatic lipid accumulation (H&E staining)	[67]
**Circulating lipids**				
Aq MO	Obese (VHFD)	Plasma	↓Cholesterol	[28]
Et MO	*Diabetic*	Plasma	↓TG and LDL-C	[62]
Aq MO	Diabetic	Serum	↓TG. No effect on TC	[47]
Me MO	Diabetic	Serum	↓ TG, total and LDL-C and ↑ HDL-C	[55]
Niazirin (seed)	*Diabetic*	Plasma	↓ LDL-C, TG and NEFA and ↑ HDL-C. ↓TC (high dose only).	[67]
Aq MO	Diabetic (2 models: HFD and STZ)	Serum	↓ TC, TG, VLDL-C, and LDL-C. ↑ HDL-C Smaller level of restoration of lipid profile in STZ- diabetic than in HFD-diabetic	[43]
**Histopathology and organ functionality**				
Et MO	Diabetic	Kidney	Restored histopathological damage in renal tissue	[62]
Aq MO	Diabetic	Serum	↓ GOT and GPT enzyme.	[43]
Aq MO	Diabetic	Pancreas and liver	Prevented histoarchitectural changes	[46,47]
MOP (seed)	Diabetic	Kidney and pancreas	Restored the normal histology of kidney and pancreas	[42]
Niazirin (seed)	Diabetic	Liver	Restored NAFLD score and hepatocyte structure.	[67]
Me MO	Diabetic	Heart	Improved histopathology	[59]
Me MO	Diabetic	Liver	Improved histopathology, ↓ GOT and ALP	[58]
Me MO	Diabetic	Kidney	Improved histopathology	[57]
Me MO (pods)	Diabetic	Pancreas	MO reversed the histoarchitectural damage of islets cells	[66]
**Gene expression of lipid metabolism and glucose metabolism**				
Aq MO	Obese (VHFD)	Liver	↓ Lipogenic proteins (FAS, SREBP1 and FSP27) and ↑ lipolytic ATGL. ↓ G6Pase	[28]
Fermented MO leaf	Obese (HFD)	Liver Muscle	↓ ACC, FAS and SREBP-1⇒ Downregulated lipogenic genes. No effect on C/EBPα, PPAR-γ and LPL. ↑ CD36, ACOX1, ATGL, HSL⇒↑ Lipid uptake, oxidation and lipolysis. No effect on CPT1, PPARα. ↑ pAMPK/AMPK. ↓BIP and PDI (muscle) ⇒ ↓ endoplasmic reticulum stress	[54]
Aq MO	Diabetic	Liver	Normalized gene expression: ↑ GS, ↓PC and caspase 3	[46]
Niazirin (seed)	Diabetic (db/db)	Liver	↑ HK and PK enzymes (glycolytic) and ↓ G6Pase and PEPCK (gluconeogenic). ↑ PPAR-α, ↓ SREBP-1 and FAS expression. ↑ ratio P-ACC/ACC.	[67]

Extracts are from leaves unless specified in brackets; MO: *M. oleifera*; ME MO: methanolic extract of MO; FM: Fermented MO leaves; NFM: Non-fermented MO leaves; Et MO: Ethanolic extract of MO; Aq MO: Aqueous extract of MO; MOP: *M. oleifera* powder; HFD: High-fat diet; VHFD: Very-high-fat diet; H&E: Hematoxylin and eosin; TC: Total cholesterol; LDL-C: Low-density lipoprotein cholesterol; HDL-C: High-density lipoprotein cholesterol; TG: Triglyceride; NEFA: Non-esterified fatty acids; STZ: Streptozotocin; GOT: Aspartate aminotransferase; GPT: Alanine aminotransferase; NAFLD: Non-alcoholic fatty liver disease; FAS: Fatty acid synthase; SREBP-1: Sterol regulatory element-binding protein; FSP27: Fat-specific protein 27; ATGL: Adipose triglyceride lipase; ACC: Acetyl-CoA carboxylase; EBPα: Enhancer-binding protein alpha; PPARα: Peroxisome proliferator-activated receptor alpha; LPL: Lipoprotein lipase; CD36: Cluster of differentiation molecule 36; ACOX1: Peroxisomal acyl-CoA oxidase 1; ATGL: Adipose triglyceride lipase; HSL: Hormone-sensitive lipase; AMPK: 5´-AMP activated protein kinase; pAMPK: phosphorylated AMPK; BIP: Binding immunoglobulin protein; PDI: Protein disulfide isomerase; GS: Glycogen synthase; PC: Pyruvate carboxylase; HK: Hexokinase; PK: Pyruvate kinase; G6Pase: Glucose-6-phosphatase; PEPCK: Phosphoenolpyruvate carboxykinase. ↑: Higher; ↓: Lower.

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
