# Peer review of "Potential of Moringa oleifera to Improve Glucose Control for the Prevention of Diabetes and Related Metabolic Alterations: A Systematic Review of Animal and Human Studies"

_nutrients, 2020, doi:10.3390/nu12072050_

Round 1

Reviewer 1 Report

The systematic review of Nova et al. summarized animal and human studies that have investigated the effects of Moringa oleifera on glucose metabolism in diabetes and related metabolic abnormalities. While the authors peformed a comprehensive research and a well conducted systematic review based on the PRISMA statement guidelines, there are some major and minor issues that should be addressed.

Major comments:

  1. The title is misleading. It is a systematic review. Please specify this and add information that it consists of animal and human studies, and not only mention "in vivo" studies.
  2. Methods and Conclusion in Abstract is missing.
  3. Lines 54-57: Some values of components of Moringa oleifera should be added to assess if Moringa oleifera is for example a protein-rich or fibre-rich plant etc.
  4. Line 61: Please add examples of proinflammatory mediators.
  5. Sometimes it is not clear if blood glucose levels were measured in fasted state or fed state (e.g. paragraph 3.1).
  6. Table 1 is very huge. Please structure table like description in the text and add subheadings within the table for example "studies with raw MO; studies with MO aqueous extract or studies with methanolic or ethanolic extract".
  7. Doses of Moringa oleifera in animal and human studies can be discussed in more detail. Are these pharmacological or physiological dosages? Furthermore, final remarks about raw material or extracts (aqueous or organic) should be drawn.
  8. What is the molecular mechanism underlying the effect how Moringa oleifera could control glucose levels based on animal studies?

Minor comments:

  1. Lines 18,19: Please correct words into "anti-hyperlipidemic", "anti-oxidative", "immunmodulatory"
  2. Figure 1: Please correct "mehodological" into "methodological" and add abbreviations in the figure legend.
  3. Table 1: Please change "measures" to "measurements", "disfunction" to "dysfunction" and "acarbosa" to "acarbose".
  4. Table 1, reference 41: How was Moringa oleifera administered?
  5. Table 2, reference 38: From which tissue was histopathology performed?

Reviewer 2 Report

This manuscript was aimed to summarize the in vivo studies of Moringa oleifera for the prevention of diabetes and related metabolic alterations. They described the general information about this plant and how it has been used as food and as medicine. Then the authors summarized the effect of this plant on glucose control shown in animal studies and human studies. In addition, they provided information about the effect of this plant on metabolic alterations related to diabetes. However, there are numerous major and minor issues in this paper.

Major compulsory Revisions

  • Generally, the in vivo study means experiments conducted using animal models, and human studies are not categorized into the in vivo study. The authors should change the title since they included human studies.
  • Organize the information in session 3. Session 3.1 described the studies using different parts of the plant, however, the following sessions summarized by different preparation methods. The authors should consider organizing the information using the same way.
  • Miss a lot of references through the paper.

Minor Essential Revisions

  • Delete the subtitle “2.1 Search strategy and selection criteria”
  • Move the “Ref” to the last column in all tables
  • Change the subtitle of “Acute/Long-term studies” to “Acute/Long-term effects”

Round 2

Reviewer 1 Report

The authors have considerably improved the manuscript and addressed all my comments.

Reviewer 2 Report

The authors have addressed all the issues raised in the original review, and I have no more comments.